# Mechanisms of Diseases Associated with Mutation in GJC2/Connexin 47

**DOI:** 10.3390/biom13040712

**Published:** 2023-04-21

**Authors:** Charles K. Abrams

**Affiliations:** Department of Neurology and Rehabilitation, University of Illinois at Chicago College of Medicine, Chicago, IL 60612, USA; cabrams1@uic.edu; Tel.: +1-312-355-5859; Fax: +1-312-996-4169

**Keywords:** *GJC2*, Connexin 47, leukodystrophy, Pelizaeus Merzbacher like disease, PMLD, PMLD1, gap junction, SPG44, HLD2

## Abstract

Connexins are members of a family of integral membrane proteins that provide a pathway for both electrical and metabolic coupling between cells. Astroglia express connexin 30 (Cx30)-*GJB6* and Cx43-*GJA1*, while oligodendroglia express Cx29/Cx31.3-*GJC3*, Cx32-*GJB1*, and Cx47-*GJC2*. Connexins organize into hexameric hemichannels (homomeric if all subunits are identical or heteromeric if one or more differs). Hemichannels from one cell then form cell-cell channels with a hemichannel from an apposed cell. (These are termed homotypic if the hemichannels are identical and heterotypic if the hemichannels differ). Oligodendrocytes couple to each other through Cx32/Cx32 or Cx47/Cx47 homotypic channels and they couple to astrocytes via Cx32/Cx30 or Cx47/Cx43 heterotypic channels. Astrocytes couple via Cx30/Cx30 and Cx43/Cx43 homotypic channels. Though Cx32 and Cx47 may be expressed in the same cells, all available data suggest that Cx32 and Cx47 cannot interact heteromerically. Animal models wherein one or in some cases two different CNS glial connexins have been deleted have helped to clarify the role of these molecules in CNS function. Mutations in a number of different CNS glial connexin genes cause human disease. Mutations in *GJC2* lead to three distinct phenotypes, Pelizaeus Merzbacher like disease, hereditary spastic paraparesis (SPG44) and subclinical leukodystrophy.

## 1. Introduction

Connexins are members of a family of integral membrane proteins, almost all of which oligomerize to form gap junctions (GJs). Gap junction channels provide a pathway for both electrical and metabolic coupling between cells [1]. GJ coupling of glial cells creates a glial syncytium [2], allowing propagation of signals such as Ca^2+^ waves (see Scemes and Giaume [3] for more detail), or potentially underlying handling potassium generated by neural activity [4,5,6,7]. Connexin hexamers can also form functional membrane hemichannels, also called connexons, in the plasma membrane; these mediate cellular processes, including metabolite influx and efflux [8,9], ATP release and signaling [10,11,12], glutamate release [8,12], glutathione release, [13] and responses to spinal cord injury [14]. Homomeric hemichannels contain six identical connexin subunits and in heteromeric hemichannels at least one connexin differs from the others. Two opposed apposed hexameric hemichannels form a dodecameric cell-cell or gap junction channel. In homotypic cell-cell channels the two hemichannels are identical, while in heterotypic channels the two hemichannels differ.

There are at least 21 members of the human connexin family and at least 20 in mice [15]. A number of these, including connexin 26 (Cx26)/GJB2, Cx29/Cx31.3/GJC3, Cx30/GJB6, Cx32/GJB1, Cx36/GJD2, Cx43/GJA1, Cx45/GJC1 and Cx47/GJC2 are expressed in the central and/or peripheral nervous system and mutations in a number of these cause central and/or peripheral nervous system disease. For example, mutations in Cx43 are associated with oculodentodigital dysplasia [16,17], a pleiotropic disorder that includes variable neurological manifestations such as seizures, cognitive deficits, and white matter abnormalities on imaging, [18,19,20,21], and mutations in Cx32 cause the most common form of X-linked Charcot Marie Tooth disease (CMTX1). In this review we will focus on the role of mutations in Cx47/GJC2 in several different phenotypes including a severe early onset dysmyelinating disorder, Pelizaeus-Merzbacher-like disease (PMLD1 or HLD2) [22], a milder, later onset disorder, hereditary spastic paraplegia (SPG44) [23], and a recently described subclinical leukodystrophy [24]. Recent general reviews of ODDD are available [25,26,27]; readers interested in CNS manifestations of Cx43 mutations may also find two older reviews [28,29] of use. Cx32 associated neurologic disease have also been recently reviewed [30,31]. All in all, at least 11 different connexins are associated with a large number of human disease phenotypes [32,33,34,35].

## 2. Glial Connexins

Astrocytes (As) express Cx30 and Cx43, and we and others have demonstrated that these connexins form homotypic (Cx43/Cx43 and Cx30/Cx30) but not heterotypic (Cx43/Cx30) GJs in transfected cells [36,37]. (The nomenclature connexin A/connexin B refers to a cell pairing configuration where cell 1, expressing connexin A is paired with cell 2, expressing connexin B to produce a gap junction channel. Where junctional voltage is applied, the voltage stated is applied to cell 2 with respect to cell 1.) In mouse hippocampal slices, the deletion of *Gja1* (encoding Cx43) alone results in about a 50% reduction in astrocytic coupling [38,39], while knockout of both Cx43 and *Gjb6* (encoding Cx30) results in the complete loss of A/A coupling [39]. Histological approaches also provide evidence that A/A GJs are comprised of homotypic Cx30/Cx30 and Cx43/Cx43 channels [40,41,42,43]. Astrocytic Cx43-positive GJ plaques are found throughout the brain while in gray matter Cx30-positive GJ plaques are more prominent [41,43,44]. Some astrocytes may also express Cx26, but some contradictory data have been presented [45]. The contribution of Cx26 to A/A coupling is unclear given the complete loss of A/A coupling seen in the Cx43/Cx30 double knockout [39]. Mutations in *GJB2* (encoding Cx26) and *GJB6* (encoding Cx30) have been noted in patients with hearing loss [46] and skin diseases [47] but are not associated with CNS abnormalities in humans or mice.

Oligodendrocytes (Os) express Cx29/Cx31.3, Cx32, and Cx47 [4,43,48,49,50]. Both Cx32 and Cx47 form functional homotypic gap junctions [36,37,51,52]. However, coupling between Cx32 and Cx47 homomeric hemichannels has not been demonstrated [36,37]. In mice, Cx29 does not form gap junctions and is localized to the adaxonal membrane of CNS myelin sheaths, where it forms hexameric rosettes [50,53,54,55]. In transfected cells, Cx29 [56] and Cx31.3 [57] do not form functional cell-cell channels, but the human orthologue, Cx31.3, appears to form hemichannels [57]. In most cells the expression of connexins leads to formation of gap junctions between different cells. However, in both oligodendrocytes in the CNS [4] and Schwann cells in the PNS [58], Cx32 is localized to the noncompact myelin where it is presumed to form reflexive gap junctions, connecting adjacent loops of myelin and shortening the pathway for diffusion between the ab- and adaxonal compartments. Experiments in rodents have also shown that Cx47 and Cx32 are found in GJ plaques that are localized on the cell bodies of oligodendrocytes [50]. Based on both conventional electron microscopy (EM) [59,60] and freeze replica immune labeling (FRIL) [5,42], it was concluded that O/O coupling did not occur to any significant degree. However, more recently two groups have reported functional O/O coupling in the mouse corpus callosum [61,62]. This O/O coupling is lost in mice that lack both Cx32 and Cx47. In addition, EM of the corpus callosum demonstrated morphologic gap junctions between oligodendrocytes [61].

We have recently examined whether Cx32 and Cx47, which are both expressed in oligodendrocytes and show substantial overlap in expression pattern in the CNS [43,49,50], can form functional heteromeric channels. Here we present some of our recently published data [63], underlying our conclusion that this is not the case. We first paired cells expressing Cx47 and Cx32 (Cx32+Cx47) or Cx32 or Cx47 alone with cells expressing either Cx30, Cx32, Cx43 or Cx47. We found that the strength of coupling for the Cx32+Cx47/Cx32 pairing was similar to that of the Cx32/Cx32 pairings and the strength of the Cx32+Cx47/Cx47 pairing was similar to the coupling produced by the Cx47/Cx47 homotypic pairing. A hallmark of homotypic gap junctions is that they produce a symmetric normalized steady state conductance voltage (G_j_-V_j_) relationship when examined by dual whole cell patch clamp recording. As shown in Figure 1a,b the G_j_-V_j_ plots for the Cx32+Cx47/Cx32 and Cx32+Cx47/Cx47 are symmetric and overlap almost perfectly with the Cx32/Cx32 and Cx47/Cx47 G_j_-V_j_ plots. This is the expected result if Cx32+Cx47 heteromers are not contributing to the cell-cell channels. Similarly, in Figure 1c,d, the Cx32+Cx47/Cx43 and Cx32+Cx47/Cx30 G_j_-V_j_ plots are asymmetric since they represent heterotypic channels. These G_j_-V_j_ plots are superimposable on those produced by Cx32/Cx47 and Cx32/Cx30 cell-cell channels, suggesting that there is no contribution from Cx32+Cx47 heteromers. Additional physiologic data arguing against heteromer formation are available [63]. Since an overlapping expression pattern is a prerequisite for heteromer formation, we also examined whether localizations of Cx32 and Cx47 were correlated when exogenously expressed in the same cells. When we performed this experiment using Cx32 and Cx30, two closely related connexins predicted to show heteromeric interactions (Koval et al., 2014), a relatively high degree of overlap in the pattern of staining was seen, as evidenced by a Pearson correlation coefficient of almost 0.5 (Figure 2). On the other hand, when Cx32WT (wild-type) was co-expressed with Cx47WT, little overlap is seen and the average correlation coefficient was very close to 0.0 (Figure 2). Figure 3 shows examples of confocal planar images used to do the analysis shown in Figure 2. (See Abrams et al. 2021 [63] for further details). In summary, the bulk of the evidence suggests that wild-type Cx47 and Cx32 do not form functional heteromers.

Connexins also underlie oligodendrocyte/astrocyte coupling. Our work in transfected cell lines shows that Cx43/Cx47 and Cx30/Cx32 (but not Cx43/Cx32 or Cx30/Cx47) form morphologic and functional GJs that have distinct electrophysiological properties, [36] but using a different approach, another group [37] showed that Cx30 and Cx47 can form functional channels. However, data showing that loss of Cx30 has no effect on localization of Cx47 in mouse brain suggest that Cx47 and Cx30 do not interact in vivo [67]. It is worth pointing out that data from Abrams et al. 2021, including that shown in Figure 1 also support our conclusion that Cx30 and Cx47 do not form functional heterotypic channels. Our finding that the Cx32+Cx47/Cx30 and Cx32/Cx30 pairs are indistinguishable, argues against both a heterotypic interaction between Cx47 and Cx30 and a heteromeric interaction between Cx32 and Cx47. Anatomical studies [4,39,41,43] make a strong case that A/O coupling consist predominantly of Cx43/Cx47 and Cx30/Cx32 heterotypic channels. (There is no evidence for a role for Cx26 in A/O coupling). A/O coupling is weak [37] or absent [61] in the corpus callosum. In the cerebral cortex, A/O coupling is dependent on Cx47, but not on Cx32, indicating that Cx43/Cx47 channels (and not Cx30/Cx32 channels) are required for A/O coupling in mouse cortex [61].

The *GJC2* gene contains two ATG codons at the upstream end of the coding region, separated by two intervening codons. Arguments have been made for assigning either the upstream [68] or the downstream [69] ATG as the start codon, but whether one of the two or both are used in vivo remains unresolved. In a recent study comparing the function of a form of Cx47 containing the longer (M1 and M4) and the shorter (M4) N-terminal domain, Fasciani and colleagues [70] show that the Cx47(M1 and M4) shows permeability differences between a positive and negatively charged dye, while the Cx47(M4) is equally permeable to both. Cx32/Cx30 and Cx43/Cx47 channels show permeability and rectification properties that may have an impact on their function in O/A coupling. For example, we have shown that Cx32/Cx30 channels are highly permeable to the monovalent anionic dye Alexafluor 350 but permeability to Lucifer Yellow (a divalent anionic dye) was not demonstrable. On the other hand, Cx47(M4)/Cx43 channels were equally permeable to both [36]. These differences in permeability may affect which signaling molecules or metabolites are able to traverse O/O and O/A junctions. Furthermore, O/A channels (Cx32/30 and Cx47(M1 and M4)/Cx43) show rectification with instantaneous (open channel) current increasing when the cell expressing Cx32 or Cx47(M1 and M4) is more positive. Fasciani et al. [70] suggest that this rectification may act to reduce potassium movement from oligodendrocyte to astrocyte during periods of neural activity where the same voltage gradient (positive in oligodendrocyte relative to astrocyte) that causes potassium to flow from O to A also leads to a reduction in open channel current and potassium flux for both Cx32/Cx30 and Cx47(M1 and M4)/Cx43 channels. They suggest that this in turn could result in increased local redistribution among coupled oligodendrocytes. It is worth noting that these considerations would not apply to Cx47(M4)/Cx43 channels where little or no rectification is seen [69,70].

## 3. Mouse Models for the Study of Glial Connexins

Genetically modified mice with targeted deletion or mutation of oligodendrocyte connexins have contributed to our understanding of the roles of these connexins in the CNS. Cx32 (*Gjb1*) null (KO) mice develop a demyelinating peripheral neuropathy after 3 months of age [71,72], with axonal changes evident by two months, [73] but only subtle changes in CNS myelin thickness are seen [74,75]. However, alterations in the expression of Cx32 in the CNS may cause a predisposition to increased intensity of response to inflammatory stimuli. For example, the Cx32KO mouse is somewhat more susceptible to EAE than the WT [76,77], and a mouse in which the Cx32 p.T55I transgene is expressed on a Cx32KO mouse background show greater susceptibility to LPS than does the WT or the Cx32KO [78]. However, to date, no baseline CNS behavioral abnormalities have been described in the Cx32KO mouse. Similarly, Cx47KO mice show no overt behavioral abnormalities. While one model shows no baseline pathological changes, [49] a different mouse model does show minimal, regionally limited myelin abnormalities [48]. Of note, loss of Cx47 has been shown to increase susceptibility to EAE [77,79] and loss of Cx32 increases susceptibility to both EAE and LPS mediated neuroinflammation [76,77,78]. The basis for the susceptibilities of both the Cx32KO and Cx47KO mice has not been fully clarified, but may be related to increased blood brain barrier permeability in these mice. Both Cx43 and Cx47 have C-terminal PDZ-binding domains and bind ZO-1, one of the defining members of PDZ family [80,81,82,83,84,85]. In Cx47KO mice, ZO-1, ZO-1–associated nucleic acid binding proteins (ZONAB), and multi-PDZ domain protein 1 (MUPP1) [86] lose their Cx47 dependent junctional localization. The resulting nuclear translocation of ZONAB, a Y-box transcription factor, may repress ErbB2 [87], a tyrosine kinase receptor implicated in oligodendrocyte development, differentiation, and survival [88,89,90]. ZONAB also regulates a number of cell cycle regulatory genes [91,92]. Thus, appropriate localization of Cx47 in the oligodendrocyte may be important for stabilization/organization of the oligodendrocyte membrane junctional complex, regulation of gene expression and control of the cell cycle.

Mice lacking both Cx32 and Cx47 show a floridly abnormal phenotype that includes abnormal movements, seizures, and death by around six weeks of age. CNS pathology includes demyelinated and remyelinated axons, enlarged extracellular spaces separating the axon from its myelin sheath, and significant oligodendrocyte apoptosis. These findings, as well as those described below, have led to the speculation that this vacuolization might be due to increased periaxonal accumulation of potassium due to failure of connexin dependent spatial buffering in the double KO [49]. Other data supporting the potassium buffering hypothesis include: (1) targeted deletion of. Kir4.1 [93], an inwardly rectifying K^+^ channel [94,95], leads to vacuolization of white matter similar to that seen in the Cx32-Cx47 dKO [93]; (2) the null allele of Kir4.1 genetically interacts with the null alleles of Cx32 and Cx47,suggesting that these three molecules are active in a common pathway [7]; and (3) reduced retinal ganglion cell activity reduces optic nerve vacuoles in the Cx32-Cx47 double knockout mice while increased activity causes increased vacuolization [7].

Cx29 (the mouse ortholog of human Cx31.3) is widely expressed in the CNS during development [96], and is found small myelinated fibers of mature brain and spinal cord [50,53,55,83]. No CNS phenotype, including normal brainstem auditory evoked potentials, was noted in mice with targeted ablation of Cx29 [97], though another group found that peripheral auditory pathways are affected [98]. A number of variants in *GJC3* have been identified in patients with deafness [99,100,101]. A few of these variants have been evaluated in exogenous expression systems and show alterations in trafficking (p.W77S) [102], channel function (p.R15G and p.L23H) [103,104], or channel structure by cryo-EM (p.R15G) [103]. However, to date no human variant of *GJC3* has been listed in ClinVar (version 20230306) [105] as pathogenic or likely pathogenic. Thus, whether *GJC3* mutations are a cause of nonsyndromic deafness requires further clarification [106].

Mice with global knockout of Cx43 are not viable due to cardiac defects [107]. Astrocyte specific knockout of Cx43 (Cx43_Astro_) does not cause overt behavioral changes but does lead to poorer performance on behavioral testing (e.g., water maze, plus maze, open field activity) [108], 50% reduction in A/A coupling [39], and increased velocity of spreading depression [109]. Mice lacking Cx30 [110] also show subtle behavioral abnormalities on detailed testing (e.g., spatial memory test and open field test). Mice with global loss of Cx30 and targeted deletion of astrocyte Cx43 showed no A/A coupling and reduced hippocampal K^+^ buffering [39], as well as motor abnormalities including increased falls and crossing time on balance beam and reduced latency to fall on rotorod testing [111]; life expectancy was not notably reduced, [39,111] indicating that loss of A/O and O/O coupling has greater impact on CNS function than does loss of A/A and A/O coupling. In contrast to the modest effects on behavior and normal life expectancy, Lutz et al. [111] found prominent white matter abnormalities in Cx43_Astro_-Cx30 dKO mice that are similar to those noted in Cx32-Cx47 dKO and Kir4.1 KO mice. In addition, work by Rouach et al. utilizing the Cx30-Cx43_Astro_ dKO mouse demonstrates that A/A channels play a role in activity dependent intercellular trafficking of potassium [112].

Several other CNS connexin double knockout mice have been examined. Magnotti and colleagues [113] investigated the Cx32-Cx43_Astro_ dKO mouse. These mice show white matter vacuolation in the corpus callosum, anterior commissure, and cerebellum as well as progressive loss of cerebellar astrocytes without loss of oligodendrocytes or microglia. They also exhibit increased foot slippage on balance beam by 6 weeks, seizures between 8 and 11 weeks and early mortality with ~85% dying before 20 weeks of age. Using a different promotor (mouse GFAP) than that used for the Magnotti studies (hGFAP) [113] to activate the Cre-recombinase to achieve astrocyte specific knockout, May et al. [114] found activated microglia and reactive gliosis in the corpus callosum and cingulum but not cerebellum, and reduced numbers of Olig2+ cells in the cingulum of Cx32-Cx43_Astro_ mice. They also found that loss of astrocytic Cx43 led to reductions in Cx47 protein levels without changes in Cx47 mRNA suggesting that Cx43 had posttranslational effects on Cx47 localization and stability.

The findings in the Cx47-Cx30 dKO mice are particularly notable [115]. There is a complete loss of O/A coupling but maintenance of O/O coupling. About 40% develop severe motor impairments and die between 42 and 90 days, but the balance live relatively normal spans and have milder motor impairment and action tremors. After p45, all mice showed vacuolation and reduced numbers of myelinated fibers in both the cortex and cerebellum. In addition, the more severely affected mice showed focal areas of tissue damage, gliosis and increased microglial staining. This variability remains unexplained but may reflect variability in the genetic background since the mice used were not fully backcrossed (~85% C57Bl/6) or be due to uncontrolled environmental factors. In contrast, Cx47-Cx43_Astro_ double knockouts show no pathology and have a normal life span [113].

Thus, the order of phenotypic severity of double connexin knockout mice seems to be Cx47/Cx32 > Cx47/30 ≈ Cx32/Cx43 > Cx43/Cx30 ≫ Cx47/Cx43 with no data in the literature about Cx32/Cx30 double knockout. This suggests that loss of all O/O and O/A coupling may have severe developmental effects not seen when some element of O/O coupling is preserved. In addition, complete loss of A/A coupling in itself is apparently less severe than when accompanied by loss of one of the connexins mediating O/A and/or O/O coupling, and partial preservation of O/O, O/A and A/A coupling seems sufficient to prevent the development of an overt phenotype.

## 4. Manifestations of *GJC2* Mutations

Homozygous or compound heterozygous mutations in *GJC2* lead to a number of different neurological phenotypes [31]. These include: (1) a severe early onset central nervous system dysmyelinating disorder, Pelizaeus-Merzbacher-like disease (PMLD1 or HLD2) [22]; (2) a milder, later onset disorder, Hereditary Spastic Paraplegia (SPG44) [23]; and (3) a recently described subclinical leukodystrophy [24]. PMLD is named for its similarities to X-linked Pelizaeus-Merzbacher disease (PMD), caused by mutations in *PLP1*, encoding proteolipoprotein (PLP), the major protein of CNS myelin [116]. Dominant mutations in *GJC2* also lead to hereditary (primary) lymphedema [117,118] and some variants in *GJC2* are a susceptibility factor for secondary lymphedema after breast cancer surgery [119]. The dominant inheritance pattern of *GJC2* associated lymphedema makes it likely that this disorder is due to a gain of function. There is no known overlap between mutations causing CNS disease and those causing lymphedema. Readers interested in further information on the role of *GJC2* and other connexins in lymphatic function and lymphedema are referred to a recent review on this topic [120].

## 5. PMLD (HLD2)

Pelizaeus-Merzbacher-like disease was originally conceptualized as a group of diseases causing a PMD phenotype but lacking *PLP1* mutations. However, as discussed below it is now often used to refer exclusively to severe hypomyelinating disease resulting from mutations in *GJC2*. In 2004 Uhlenberg et al. [22] identified *GJC2* mutations in 3 of 6 PMLD families examined. A few years later Henneke and colleagues [121] showed that among patients with a PMD phenotype but lacking mutations in the *PLP1* gene, about 8% were found to have biallelic mutations in *GJC2*. However, patients classified as not *GJC2* related may actually carry mutations in portions of the *GJC2* gene not sequenced in this study, such as the promotor region [122]. In addition, while significant clinical information about the patients with *GJC2* mutations is provided, there is less information about the phenotypes of the *GJC2* mutation negative patients. Some *PLP1* mutation negative patients might have had additional phenotypic features that, with our expanding understanding of the spectrum of hereditary hypomyelinating disorders, would now result in their being distinguished from those with a PMD phenotype. Thus, it appears that the term PMLD was originally used fairly broadly as a synonym for non-*PLP1* associated hereditary leukodystrophy (HLD), but this seems to have fallen out of favor. For example, the original description of patients with HLD3 due to *AIMP1*/p43 mutations classified these as PMLD. [123] However the use of the term PMLD has been questioned in this context [124,125]. A search for PMLD in OMIM only reveals *GJC2* and a single report of *SNAP29* mutations [126] (the cause of cerebral dysgenesis, neuropathy, ichthyosis, and keratoderma syndrome (CEDNIK)) as causative of PMLD. The NORD Rare disease database https://rarediseases.org/rare-diseases/pelizaeus-merzbacher-disease/ (accessed on 11 March 2023) separates PMLD (caused by *GJC2* mutations) from their list of other diseases that need to be distinguished from PMD. Nonetheless, the term PMLD has also been used recently to describe both an unusual case associated with mutations in NPC1, the gene associated with Niemann Pick disease types C1 and D [127] and a RARS mutation associated hypomyelinating phenotype [128].

Cardinal manifestations of PMLD include nystagmus within the first 6 months of life, cerebellar ataxia by 4 years, and spasticity by 6 years of age [22,129,130,131]. All patients with PMLD have extensive signal changes on MRI consistent with abnormal myelination; relative sparing of the corticospinal tracts and involvement of the brainstem white matter has been reported [22,129,130,131,132,133,134,135]. Axonal damage appears to be less prominent since MRI spectroscopy studies in two patients showed normal or near normal choline, N-acetyl aspartate and creatine levels [131]. (Abnormal N-acetyl aspartate levels indicate axonal damage.) While imaging studies are informative, our understanding of this disorder has been limited by the lack of published autopsy studies of the brains of patients with *GJC2* associated diseases.

To date, at least 45 different variants in the coding region of *GJC2* have been associated with recessively inherited CNS disease [22,24,121,129,130,131,132,133,136,137,138,139,140,141,142,143,144]. These include missense, nonsense, and frameshift mutations. Two promoter mutations c.-167A>G [122,145,146] and c. -170A>G [147] and one mutation in the 5’ noncoding region at a predicted highly conserved splice site [148] have also been identified.

The advent of the genome aggregation database (gnomAD, https://gnomad.broadinstitute.org/) (accessed on 11 March 2023) and other databases of population genetic variability have reinforced the idea that not all detected variants are likely to be pathogenic. This idea arises because some variants found in these population databases are simply too common to contribute to the disease caused by mutations in those same genes. However, to make such assertions one needs to have a reasonable estimate of the population frequency of the relevant disorder. The prevalence of PMD is thought to range from 1:200,000–1:500,000 in the US [149], with incidence possibly as high as between one [150] and 1.9 per 100,000 male live birth [151]. A recent study from China [137] suggests that PMD is roughly 10 times more common than *GJC2* associated disorders. Thus, the incidence of *GJC2* associated leukodystrophy is likely no more than 1 in 500,000 births. Thus, for a recessive disorder such as PMLD, any variant appearing with a frequency greater than about 1/7000 alleles would be highly unlikely to be pathogenic, and alleles appearing with a frequency of 1/1000 or more would have to be responsible for at least half of all PMLD. Thus, variants such as p.G146S [138] (in this review, the residue numbering scheme for *GJC2* follows that used in Orthmann-Murphy et al. 2007 [69]) and p.T395I [121] are likely not pathogenic.

## 6. Hereditary Spastic Paraparesis (SPG44)

Orthmann-Murphy et al. [23] described a large family with three affected members carrying a homozygous I33M mutation of *GJC2* [23]. Three individuals (two brothers and a cousin) had a phenotype similar to that of other patients with hereditary spastic paraparesis (HSP); *GJC2* related HSP is designated as SPG44. All three patients were ambulatory until at least the age of 30 and all three showed spasticity, hyperreflexia, and intention tremor on exam; none had nystagmus. All heterozygous individuals in the family were normal. More recently, two siblings from a second family were described with late onset ataxia, tremor, and pyramidal signs. The symptomatic family members were both found to be homozygous for the p.V271L mutation [152]. In summary, Cx47 associated HSP is a much milder than is PMLD. Consistent with this, white matter changes seen on cerebral MRI are much milder than those seen in PMLD [23].

## 7. Subclinical Leukodystrophy

At least one patient with a subclinical *GJC2* related leukodystrophy has been reported [24]. The patient, who carried a previously unreported homozygous mutation in Cx47, p.R98L, had mild diffuse white matter abnormalities on MRI imaging, but lacked clinical neurological findings.

## 8. Mechanisms of *GJC2* Associated Diseases

All CNS diseases associated with *GJC2* are inherited in an autosomal recessive pattern, suggesting that this disorder is characterized by the loss of function of *GJC2*. However, this alone would not explain the differences in severity seen in different forms of *GJC2* related disease and would be discordant with the lack of overt phenotype seen in the Cx47 knockout mouse [48,49]. (We acknowledge that a mouse model is not always able to fully recapitulate all aspects of human disease.) There are several possible reasons why some mutations in *GJC2* cause a severe, PMLD phenotype while others do not.

### 8.1. Mislocalization

Work from our lab in both cell lines and primary oligodendrocyte cultures suggests that both the HSP and PMLD mutants are unable to form normally functioning gap junctions when assessed by dual whole cell recording [23,69,153]. In addition, we found that mutations associated with PMLD lead to loss of normal subcellular localization to gap junctions while a mutation associated with the milder HSP phenotype shows no such mis-localization [23]. Studies from our lab in HeLa and Neuro2a cell lines showed that expression of WT Cx47 or the mild HSP related mutant resulted in intercellular puncta consistent with the formation of gap junction plaques, while Cx47 mutations associated with PMLD resulted in intracellular staining colocalizing with Grp94, an ER marker [69] and few (p.M283T) or no (p.P87S and p.Y269D) intercellular plaques. This suggests the possibility that localization of Cx47 to the gap junction is important independent of the effects on channel function, and that milder mutations represent a partial loss of function (no gap junctional coupling but preserved localization to plaques) while more severe mutations affect both. This hypothesis is also consistent with findings in our studies of the p.R98L variant associated with an extremely mild subclinical leukodystrophy [24]. In that case, the protein localized to intercellular plaques and showed a reduced level of homotypic coupling, while heterotypic coupling with Cx43 was not significantly reduced. The notion that preservation of normal subcellular localization leads to reduced levels of pathogenicity independent of the ability to support gap junction communication is concordant with the observations noted above regarding the importance of Cx47 in proper localization of ZO-1, ZO-1-associated nucleic acid binding proteins (ZONAB), and multi-PDZ domain protein 1 (MUPP1) [86]. However, a study by Kim et al. [68] raises the possibility that differential localization to intercellular plaques is not a feature distinguishing mutations causing more or less severe phenotypes. They examined four variants associated with PMLD and found that these did form plaques but failed to induce intercellular coupling; however, these were GFP tagged, which may have normalized the trafficking. (Unpublished work from my laboratory suggests that the addition of GFP to the C-terminus of mutant forms of Cx32 showing abnormal trafficking rescues their ability to form plaques). Two additional GFP tagged variants, p.G149S and p.T398I showed plaque formation and significant intercellular coupling; this is consistent with the possibility, discussed above, that these are not pathogenic variants.

### 8.2. Activation of the Unfolded Protein Response (UPR)

Several *GJC2* variants associated with PMLD show retention of mutant protein in the ER; this would be expected to cause ER stress and activation of the UPR pathway, but was not demonstrable in our original studies expressing PMLD mutants in HeLa cell lines [69]. We [153] then expressed CX47WT and mutants in primary oligodendrocytes lacking endogenous Cx47; in this system, the three PMLD associated mutants (p.P87S, p.Y269D and p.M283T) show ER retention (see Figure 4) and evidence of activation of the UPR and apoptotic pathways. On the other hand, the milder SPG44 associated mutation (p.I33M) shows a wild-type-like subcellular distribution and no activation of the UPR or apoptotic pathways. Using exogenous expression in MO3.13 oligodendrocyte cell lines, Chen and colleagues have demonstrated that two additional PMLD associated mutations, p.P70S and p.I43M, also triggered ER stress [154].

### 8.3. Differential Interactions with Other Connexin Isoforms

Another possibility is that in *GJC2* related disorders, interactions between mutant forms of Cx47 and wild-type Cx32 may contribute to the disease phenotypes. For example, the greater severity of mutations causing the more severe PMLD phenotype might be due to abnormal interactions with Cx32, not seen with Cx47WT or the mutation causing the milder HSP phenotype. Such interactions might produce a situation where both Cx32 and Cx47 function was reduced creating a severe phenotype such as that seen in the Cx32-Cx47 dKO mouse described above [48,49]. To examine the possibility that differential heteromeric interactions between Cx32 and the p.I33M, p.P87S, p.Y269D, and p.M283T mutants of Cx47 might explain the different severities of disease caused by these mutations, we examined pairs of cells wherein one cell was expressing both Cx32 and one of these Cx47 mutants, and the other cell expressed Cx30 or Cx32 alone [63]. We saw no evidence of a quantitative or qualitative effect of the expression of the mutant on the magnitudes of coupling or the normalized conductance voltage relations produced by these channels. Colocalization of Cx32 and Cx47 mutants would be a prerequisite for the formation of heteromeric channels. Therefore, we examined the distribution of Cx32 and a mutant form of Cx47 when co-expressed in cell pairs. As shown in Figure 2, and noted above, when we expressed Cx32 and Cx30 in the same cell, these two closely related connexins showed significant overlap in staining. (Pearson correlation coefficient was almost 0.5). This finding is consistent with the known heteromeric interactions between Cx30 and Cx32 [155]. When Cx32WT was co-expressed with Cx47WT the correlation coefficient was close to 0, while three of the mutants showed a slight, non-significant negative correlation with Cx32; however, the p.Y269D mutant showed a significant negative correlation with Cx32, suggesting that the overlap of this connexin with Cx32 is less than that predicted by chance. Examples of planar confocal images representative of those used for these studies are shown in Figure 3.

## 9. Conclusions

This review highlights the fact that we still have much to learn about the roles of connexins in the central nervous system in normal and pathological states. In particular, I have focused on the role of mutations in *GJC2*, the gene encoding Cx47, in hypomyelinating diseases of the CNS. There is still much to learn about these disorders, and progress will likely require the development of more robust cellular or animal model systems for the study of these diseases. As noted above, the Cx47 KO mouse has a very mild phenotype, and the phenotype of the only current PMLD “knockin” mouse is also not very robust. It is likely that further progress in our understanding of these disorders will require the use of other model systems such as differentiated human stem cells expressing the relevant mutant proteins. The establishment of such a system should also allow for better evaluation of treatments for these disorders.

## Figures and Tables

**Figure 1 biomolecules-13-00712-f001:**
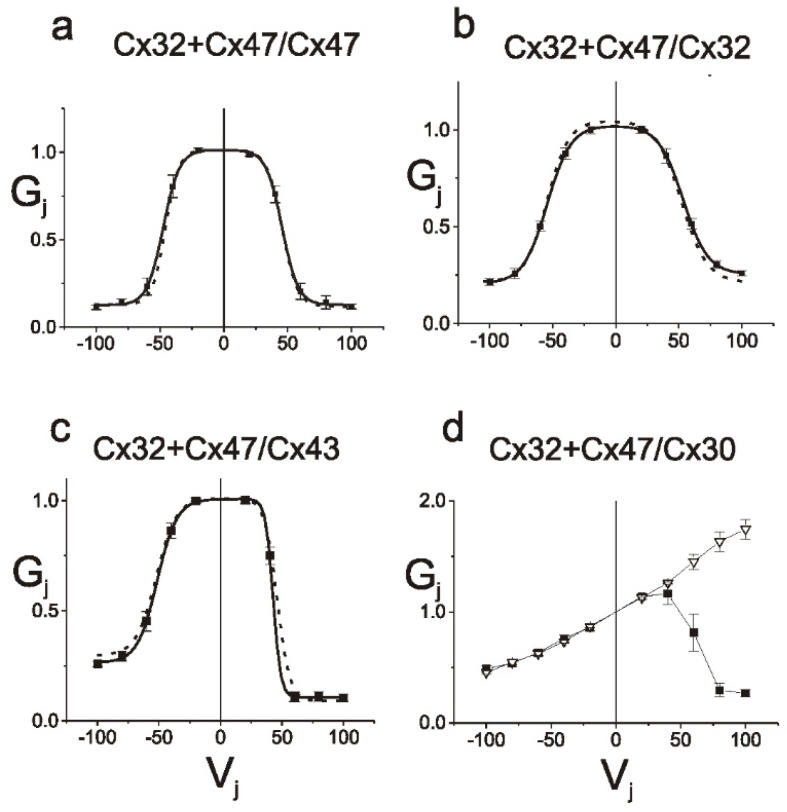
Properties of junctions between cells expressing Cx32 and Cx47 paired with cells expressing (**a**) Cx47, (**b**) Cx32, (**c**) Cx43 or (**d**) Cx30 alone. Average normalized steady state G_j_-V_j_ relations for Cx32+Cx47/Cx47, Cx32+Cx47/Cx32, Cx32+Cx47/Cx43 and Cx32+Cx47/Cx30 pairings are shown. The average normalized steady-state (filled squares) junctional conductance (G_j_) at each V_j_ was calculated as described [63]. The solid lines are the best fit of the data to a double Boltzmann distribution as described in the Methods, and the dotted lines are the best fit of data for Cx47/Cx47, Cx32/Cx32, or Cx47/43 heterotypic junctions. G_j_-V_j_ relations for Cx32+Cx47/Cx47, Cx32+Cx47/Cx32, Cx32+Cx47/Cx43, and Cx32+Cx47/Cx30 pairings are indistinguishable from those for Cx32/Cx32, Cx47/Cx47, Cx47/Cx43 and Cx32/Cx30 junctions respectively. The error bars represent SEM. Figure and legend modified from Abrams et al., 2021 [63]. Used with permission of Wiley Periodicals LLC Copyright © 2023.

**Figure 2 biomolecules-13-00712-f002:**
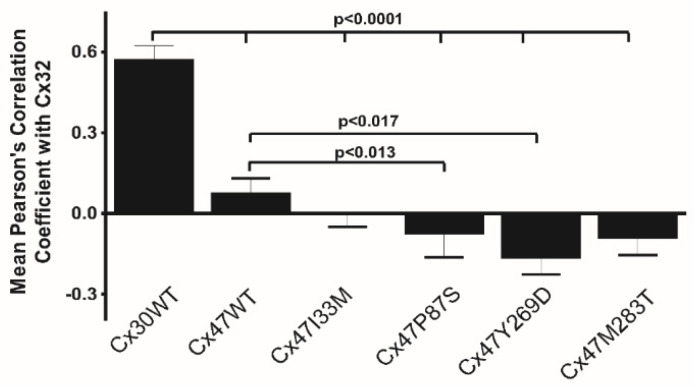
Graphic representation of results of experiments to determine colocalization of Cx32WT with Cx47WT and mutants. As shown, cells expressing Cx30WT and Cx32WT show significantly more correlation in the overlap of staining than do any of the other pairs of co-expressed connexins. Cells expressing Cx32WT and Cx47WT showed essentially no correlation of staining for the two connexins and cells expressing Cx32WT with either Cx47I33M, Cx47P87S, or Cx47M283T showed only a slight and non-significant negative correlation of staining. Cells expressing Cx47Y269D and Cx32WT did show a statistically significant negative correlation of staining. Figure and legend modified from Abrams et al. 2021 [63]. Used with permission of Wiley Periodicals LLC Copyright © 2023.

**Figure 3 biomolecules-13-00712-f003:**
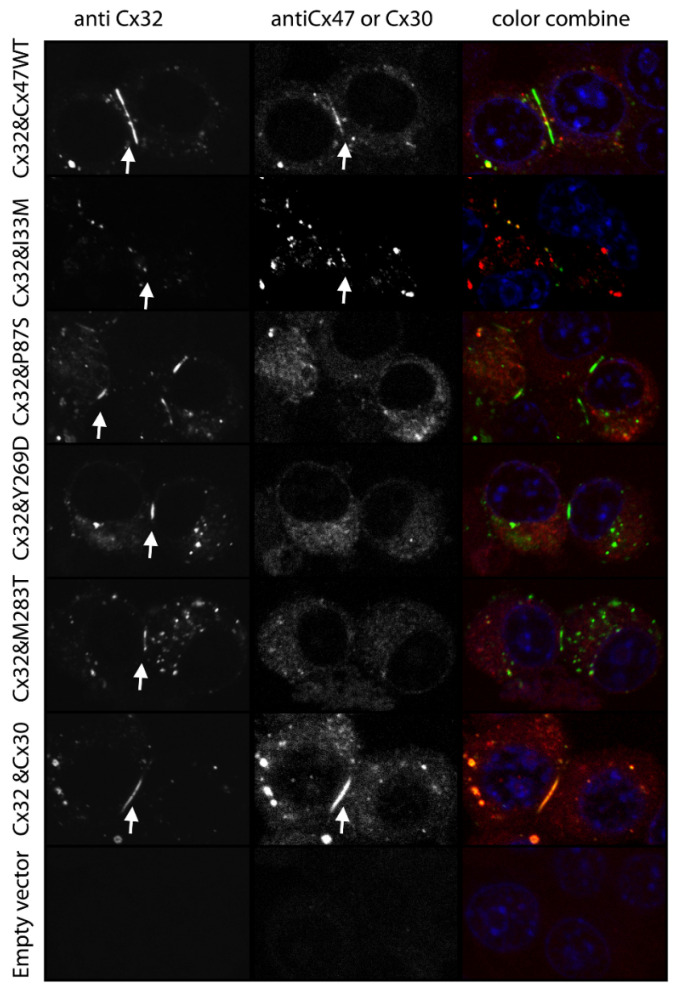
Examples of images used for determination of colocalization as summarized in Figure 2. Arrows show selected plaques for Cx30WT, Cx32WT, Cx47WT and Cx47I33M. A few non-junctional areas in Cx32+Cx47WT and Cx32+Cx47I33M expressing cells show overlap between staining for Cx32 and Cx47. Non interacting connexins can participate in the same junction; however, they are regionally distinct within that junction [64]. The areas of overlap may represent internalized (annular) gap junctions [65] which often tend to be engulfed en masse, or connexin containing organelles that are part of the degradation pathway [66]. Scale bar 20 microns. Figure and legend modified from Abrams et al. 2021 [63]. Used with permission of Wiley Periodicals LLC Copyright © 2023.

**Figure 4 biomolecules-13-00712-f004:**
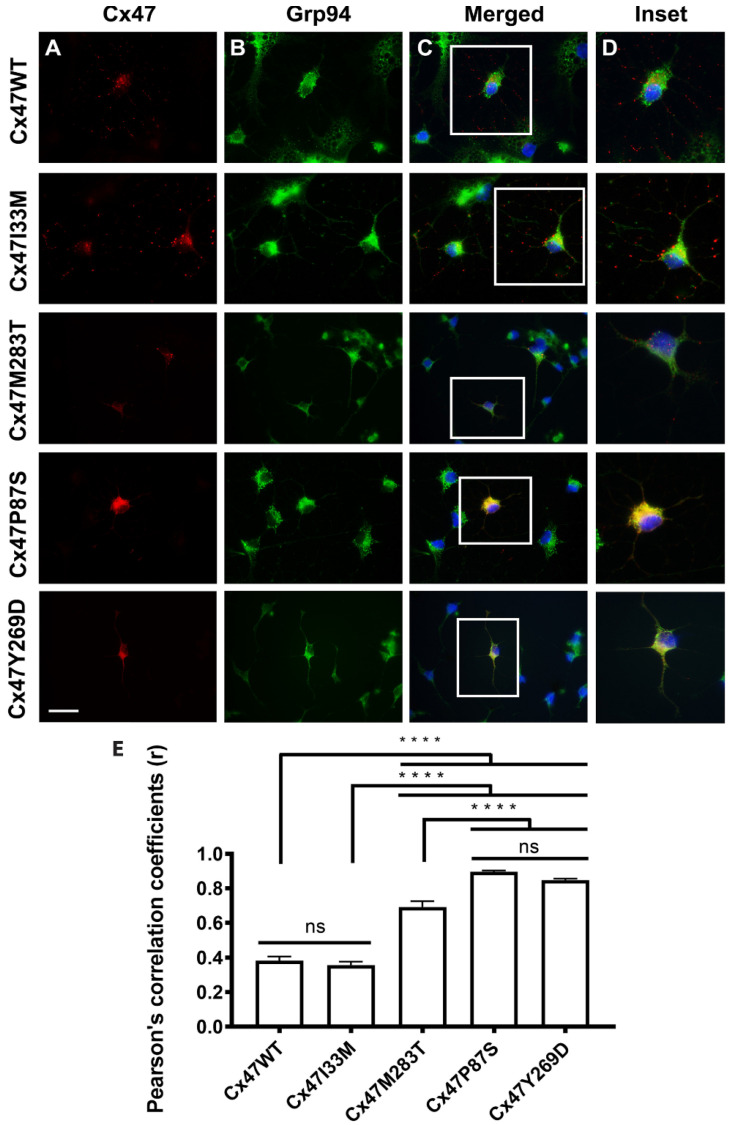
Expression of Cx47WT and mutants in primary OLs. (**A**) Cx47 staining (red) for Cx47WT and p.I33M show a punctate staining pattern while PMLD associated mutations show diffuse cytoplasmic staining (p.P87S, p.Y269D) or both punctate and diffuse staining (p.M283T). Empty vector control showed no Cx47 staining. (**B**) Staining for GRP94 (green), a marker of the ER compartment. (**C**) Staining for Cx47 (red), GRP94 (green) and DAPI (blue) shows colocalization of p.P87S and p.Y269D and slightly reduced colocalization of GRP94 with p.M283T. Scale bar = 25 μm. (**D**) Insets of indicated areas in C. (**E**) Quantitation of colocalization with Pearson’s correlation coefficient. ANOVA *p* < 0.05, Tukey post-hoc test, **** *p* < 0.0001, ns = not statistically significant. Figure and legend from Flores-Obando 2022 [153] used with permission of Elsevier. Copyright © 2023.

## Data Availability

Underlying data from the author’s lab is available upon reasonable request.

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
