# Peer review of "Mechanisms of Diseases Associated with Mutation in GJC2/Connexin 47"

_biomolecules, 2023, doi:10.3390/biom13040712_

Round 1

Reviewer 1 Report

The author presents a well organized review that focus on Cx47 constituted channels under physiological and pathological conditions.

Only very few and minor suggestions are mentioned bellow: 

I could find description of the restrictive permeability of Cx47 channels and the diffusion barrier of Cx47-Cx43 channels that is abolished by a mutation associated with leukodystrophy, the Cx47P90S. doi.org/10.1002/glia.23471.

A calibration bar is missing in Figure 3.

In the introduction line 40 it could be mentioned that Cx hemichannels also serve for metabolite influx

Reviewer 2 Report

Astroglia express Connexin 30 (Cx30)-GJA1 and Cx43-GJA1, while oligodendroglia express Cx29/Cx31.3-GJC3, Cx32-GJB6, and Cx47-GJC2. Oligodendrocytes couple to each other through Cx32/Cx32 or Cx47/Cx47 homotypic channels and they couple to astrocytes via Cx32/Cx30 or Cx47/Cx43 heterotypic channels. Astrocytes couple via Cx30/Cx30 and Cx43/Cx43 homotypic channels. As a continuation of his previous research themes, Author does focus on the role of mutations in GJC2, the gene encoding Cx47, in different phenotypes including a severe early onset dysmyelinating disorder, Pelizaeus-Merzbacher-Like disease (PMLD1 or HLD2), a milder, later onset disorder, Hereditary Spastic Paraplegia (SPG44), and a recently described subclinical leukodystrophy. Cellular and mouse (Cx47 KO or PMLD “knockin”) models were also evaluated. Author concludes that the study of these diseases will likely require the development of more robust cellular/animal model systems, such as use of differentiated human stem cells expressing the relevant mutant proteins. The establishment of such a system should also allow for better evaluation of treatments for these disorders.

Reviewer 3 Report

This review provides an updated approach to the understood mechanisms of demyelinating disorders caused by oligodendrocyte Cx47 mutations.

Comments and Suggestions 

  1. The author mentions in the Introduction that connexins can form intercellular channels and functional hemichannels; however, there is no reference to the hemichannel-forming ability of Cx47. Is the role played by Cx47 hemichannels in oligodendrocytes sufficiently well-known? Could hemichannels be involved in pathogenic mechanisms?
  2. The author should comment and discuss, in addition to the connexin composition of the O/O and A/O channels, their permeability/selectivity properties. A diffusion barrier between oligodendrocytes and astrocytes was demonstrated more than 20 years ago, and its molecular bases were recently described. What functional relevance does the presence of a directional diffusion barrier mediated by heterotypic Cx47-Cx43 channels between O/A have? Has any Cx47 mutation been described that affects the O/A barrier?
  3. This review proposes that mislocalization and activation of the unfolded protein response may be the mechanisms responsible for the variable severity of the Cx47 mutations, but what is their impact on the myelin vacuolation?
